# Plant-Based Biostimulants for Seeds in the Context of Circular Economy and Sustainability

**DOI:** 10.3390/plants13071004

**Published:** 2024-03-31

**Authors:** Hisham Wazeer, Shraddha Shridhar Gaonkar, Enrico Doria, Andrea Pagano, Alma Balestrazzi, Anca Macovei

**Affiliations:** Department of Biology and Biotechnology ‘L. Spallanzani’, University of Pavia, Via Ferrata 9, 27100 Pavia, Italy; hisham.wazeer01@universitadipavia.it (H.W.); shraddha.shridhargaonka01@universitadipavia.it (S.S.G.); andrea.pagano01@universitadipavia.it (A.P.); alma.balestrazzi@unipv.it (A.B.)

**Keywords:** agri-food by-products, bioactive compounds, biostimulants, seed priming, seed germination

## Abstract

Plant-based biostimulants (PBs), agents rich in bioactive compounds, are emerging as key players able to sustainably improve plant growth and crop productivity to address food security. PBs are generally applied as foliar spray or soil irrigation, while more recently, the application as seed priming treatments is being envisaged as a highly sustainable method to also improve seed quality and germination. Therefore, this review proposes to explore the use of PBs for the seeds industry, specifically discussing about the relevance of product market values, sustainable methods for their production, why and how PBs are used for seed priming, and pinpointing specific strengths and challenges. The collected research studies indicate that PBs applied to seeds result in improved germination, seedling growth, and stress tolerance, although the molecular mechanisms at work are still largely overlooked. The high variability of bioactive molecules and used sources point towards a huge reservoir of nature-based solutions in support of sustainable agriculture practices.

## 1. Introduction

To meet the ZERO HUNGER target SDG2 of the 2030 UN Agenda for Sustainable Development, agricultural productivity must double by 2030 (https://www.un.org/sustainabledevelopment/, accessed on 29 March 2024). This ambitious goal should be met by implementing versatile agricultural practices that increase productivity while ensuring sustainability in the food production systems, maintenance of the ecosystems, and climate resilience. Intensive farming, based on high input of chemical treatments, can no longer be considered a suitable solution since crop productivity is enhanced at the cost of severe and persistent environmental problems [1].

When considering agricultural productivity, seeds and seed quality are essential determinants, strongly linked to crop establishment. Efficient agricultural production can be attained by using high-quality seeds, which is considered a critical characteristic by the seed industry and farmers. Methods to enhance seed quality are necessary to drive an efficient crop establishment and avoid waste of seed materials due to poor quality. High-quality seeds can be defined by their vigor, expressed by natural robustness associated with rapid and complete germination under a wide range of environmental conditions [2]. Technologies known as seed priming can be applied to improve seed quality [3,4,5,6]. These are adaptable, cost-effective, and environmentally friendly approaches that rely on controlled imbibition in a pre-sowing phase. It has been shown that seed priming synchronously increases germination, improving seedling vigor, plant growth, and stress resilience [7,8,9,10,11]. Recently, plant extracts have been proposed as novel seed-priming treatments. Compounds such as pineapple bromelain [12] or garlic allelochemicals [13] have been successfully applied to enhance germination under adverse conditions.

Natural bioactive compounds of plant origin, acting as enhancers of plant growth and development, can be defined as plant-based biostimulants (PBs). Active research on PBs progressed rapidly, mainly prompted by the growing interest of seed industries and farmers who continuously search for strategies to improve crop performance and yield stability while reducing the use of synthetic agrochemicals. PBs production may be expensive, but when these are conveniently obtained from agro-industrial and urban waste according to the principle of “re-use, reduce, recycle”, the process becomes more sustainable for both society and the economy [14]. Agricultural waste is a particularly valuable source of PBs as it is abundant, cheap, renewable, and not in competition with the food/feed chain, and its valorization mitigates the impact that organic waste would have on eutrophication. According to FAO and UN Environment Programme (UNEP) statistics, more than 13% of globally produced food is lost in the supply chain after harvest, and an additional 17% loss represents household waste (https://www.fao.org/newsroom/detail/international-day-of-awareness-of-food-loss-and-waste--fao-calls-for-circular-model-in-agrifood-systems/, accessed on 29 March 2024). Turning this waste into PBs to be used as crop productivity enhancers could address food security in a sustainable approach in line with the circular economy roadmap. Different types of plant biomass waste and algae grown on urban wastewater can be repurposed for the production of natural PBs providing bioactive natural substances, like phenolic compounds, polyamines, phytohormones, amino acids, and minerals [15].

When considering the interests of the scientific community, seed priming and plant biostimulants topics are growing at a similar pace (Figure 1). The PubMed (https://pubmed.ncbi.nlm.nih.gov/, accessed on 29 March 2024) search carried out at the end of March 2024 indicates that the number of articles so far available in this database includes a total of 1233 for seed priming (Figure 1a) and 1169 for plant biostimulants (Figure 1b). It is important to underline that these numbers include only research papers, not reviews. A recently published biometric study has evaluated the trends in seed priming research during the past 30 years, evidencing a continuous increase in popularity, advancements, and significance of research, along with intense collaborations worldwide [16]. Reviews concerning biostimulants are now focusing on understanding the molecular aspects at the basis of these treatments [17]. Even though these two topics, when looked at separately, include a considerable number of articles published, if we combined them in a unique search (Figure 1c), it is possible to evidence a pronounced gap of knowledge regarding the use of biostimulants as seed priming agents. In this case, the PubMed search using “Seed Priming” AND “Plant Biostimulants” as keywords found only 43 articles of recent publication during 2016–2024 (Figure 1c). Therefore, the current review work proposes to summarize the funding observed in these articles to pinpoint aspects related to: (1) sources of PBs and bioactive molecules mostly used so far, (2) how are these obtained and applied on seeds, (3) which species are these applied to, (4) which are the effects on germination, seedling growth, and development, (5) how these effects may be explained. By proposing this review of literature, the current work proposes to bring forth recent data related to the use of PBs as seed priming agents, underlining the “knowns and unknowns” related to this subject. Moreover, when addressing issues related to sustainability and circular bioeconomy, particular care should be placed on evaluating the expected market trends, stakeholders’ needs and wants, economic feasibility, environmental impact, and competitive advantages of new technologies [18]. By conducting accurate analyses of the seed and PBs markets, these products can be put to use for new challenges in agriculture, guaranteeing more competitive markets, which creates business cases and more opportunities for jobs and growth.

## 2. Agri-Food Market Trends Regarding Seeds and Biostimulants

Today’s agriculture systems are called to face multiple challenges; on the one hand, they must be more productive, and on the other hand, they must address the growing food demand sustainably. Over the years, the constant application of agrochemicals has disrupted the natural balance of plant nutrition and health, paradoxically leading to decreasing yields and impaired plant health [19]. In light of joint global efforts to halt climate change by reducing greenhouse gas (GHG) emissions, the whole agri-food system must implement eco-sustainable innovations to confer plant stress resilience and wider climate adaptation.

The use of high-quality seeds is a proxy of the seed market, being translated not only as a means to start new cultivation with healthy, well-identified, and traced products, but also as an efficient way to transfer technology to farmers to help them cope with innovative and sustainable practices. This translates into a continuous increase in the seed market trend. For instance, the Commercial Seed Market size in 2023 was USD 61.9 billion (Bn) and is expected to rise to USD 125.2 Bn by 2033, with a 7.3% CAGR (Compound Annual Growth Rate) increase for the next 10 years (https://market.us/report/commercial-seeds-market/, accessed on 29 March 2024). For earlier periods, the Global Market Analysis, Insights, and Forecasts for 2018–2025, indicated that the commercial seed market increased from USD 40.70 Bn in 2017 to a predicted USD 61.32 Bn by 2025 (Figure 2), with a 5.8% CAGR increase (https://www.fortunebusinessinsights.com/industry-reports/commercial-seed-market-100078, accessed on 29 March 2024). When also considering the seed treatment market, this was projected to register a CAGR of 8.2% in the 2020–2025 time period. Seed treatments are successful examples of the collaboration between the seed industry and the crop protection industry, united to propose innovations for the future of modern agriculture. The use of non-chemical or bio-based seed treatment is growing at a fast rate due to the shift in preference for organic products.

Therefore, agronomical applications with new, sustainable products are predicted to register increased yields and resilience to fluctuating and challenging environmental conditions. In this view, the recycling of plant biomass and its reuse to produce PBs is highly encouraged. The PBs market was estimated at USD 2.6 Bn in 2019 and is projected to reach USD 4.9 Bn by 2025 (Figure 2), at a CAGR of 11.24% during the forecast period (MarketsandMarkets 2019, https://bit.ly/3tgd4D3, accessed on 29 March 2024). The rapid increase in PBs market trends is driven by several key factors (e.g., global food security, technological advancements, consumer awareness, regulatory support), reflecting the growing demand for sustainable agricultural practices, the need for improved crop productivity, and the rising awareness of environmental issues.

## 3. Sustainable Methods for the Production of Plant-Based Biostimulants

As we have seen so far, effective valorization of waste by-products can contribute to helping reduce environmental stress by decreasing unwarranted pollution, guaranteeing the circular economy principles and providing an opportunity for additional income for the corresponding sectors [20,21]. Recovering bioactive compounds from plant waste can contribute to significantly reducing the costs and increase the environmental sustainability of the food system. Therefore, green and sustainable separation of natural products from agro-industrial waste is clearly attractive considering both socio-environmental and economic aspects. In this part, the principal extractive techniques, including emerging green and sustainable separation approaches to obtain target compounds from agri-food by-products, are discussed.

Before plunging into typologies of extraction systems, it is important to keep in mind that food waste is usually generated in different forms and compositions, according to regional, seasonal, and processing characteristics. Moreover, the content of valuable compounds is lower compared to the initial sources (e.g., fresh fruits or vegetables). For these reasons, often the resultant processing cost may be high, with low recovery yield and revenues. Food wastes are already processed materials, which are susceptible to microbial growth and require both preservation and fast treatment [22]. Collection timing at the source is crucial to preserve the content of valuable compounds as well as reduced material transportation. Therefore, the development of an economically feasible, sustainable, and safe recovery of high added-value compounds from food waste requires a holistic approach, which includes several parameters that must be considered, such as: (1) the abundance, location, and distribution of waste material; (2) production frequency; (3) the development of a methodology that provides the highest recovery yield of different compounds and discharges minimum quantities of by-products in the environment; (4) the non-destructive separation of valuable compounds and their reutilization for other products; (5) preservation of the functional properties of each class of compounds, from source to final product; (6) obtaining a standardized concentration of target bioactive compounds with stable characteristics; (7) use of green solvents for the extraction process [22,23,24].

One of the current major research challenges is the identification and development of the best “extraction” conditions, e.g., conditions that improve the release of bioactive compounds from the plant matrix in which they are enclosed. The recovery of bioactive compounds from fruit and vegetable waste includes sample preparation, extraction, production of powdered extract (if needed), isolation, and purification by chromatography [25]. Each of these phases involves the use of different technologies, aiming at the recovery of specific molecules with distinct physicochemical characteristics. For instance, it is known that food microstructure affects the bio-accessibility and bioavailability of nutrients such as antioxidants [26]. In compliance with the sustainability criteria, each of these processes must have a reduced environmental impact both in terms of energy and use of solvents. According to the principles of circular economy, alternative and innovative extraction systems should be used, depending on the characteristics of the target compounds to be extracted. The main emerging techniques, considered innovative, rapid, low-cost and clean, are the following: (1) Green solvent extraction; (2) Ultrasound-Assisted Extraction (UAE); (3) Pulsed Electric Field (PEF); (4) Microwave Assisted Extraction (MAE); (5) Pressurized liquid extraction (PLE); (6) Supercritical Fluid Extraction (SFE); (7) Enzyme-Assisted Extraction (EAE).

The recovery of valuable compounds from food wastes usually includes the solubilization of solutes into one or more solvents that provide a physical carrier to transfer them between different phases (e.g., solid, liquid, and vapor). Since the recovered bioactive compounds are generally destined for food applications, the selected solvents should meet the following requirements: cheap and easily accessible in the food industry, generally recognized as safe, reusable and recyclable, able to inhibit enzyme activity where it is appropriate, and preserve the functional properties of target compounds [27]. Absolute ethanol meets the requirements listed above. Hydroethanolic mixtures have been widely used for the recovery of several polyphenols, whereas different phenolic fractions can be obtained based on polarity by varying the alcohol concentration in the mixture [28]. Hydroethanolic mixtures are less efficient in extracting other compounds like organic acids or complex sugars and thus additional purification steps are required. Moreover, the selection of the appropriate solvent is much more difficult in the case of lipophilic compounds, e.g., carotenoids. For these, polar aprotic (e.g., acetone) and non-polar (e.g., ethyl acetate) mediums are preferred [29], but these solvents are not food-graded and should be completely removed from the extract before its reutilization in food formulations. An important alternative is the use of hydrotropic solvents and amphiphilic organic salts, which can sparingly dissolve soluble organic compounds in aqueous solutions.

Among the innovative and green technologies, SFE and EAE are the most used and versatile. SFE is a process based on the use of solvents above or near their critical temperature and pressure to recover extracts from solid matrices. This technology uses renewable solvents, such as CO_2_, which is an ideal solvent to extract mainly non-polar compounds such as carotenoids or essential oils. If other co-solvents, such as ethanol or methanol, are added it is possible to also recover polar compounds like polyphenols or some amino acids. Solutes are extracted according to the convection and diffusion principles; after the extraction, a separation step follows, where the pressure is reduced, allowing the precipitation of the solutes [22]. Even if SFE is considered a safe and green technology, allowing the capture of CO_2_ from industrial plants nearby, the scale-up costs are still high and a major point to be considered at the industry level is energy consumption [30]. The use of additional extraction systems like microwaves, ultrasound, and enzymatic pretreatments can help to reduce the number of organic solvents used for extraction, improving the yield and the processes’ economics. On the other hand, EAE is perhaps the most environmentally and economically sustainable method of extraction. The main mechanism underlying EAE involves enzymes that degrade the plant cell wall (e.g., glucanases, pectinases, protease, pectinase, pectinesterase, cellulase, hemicellulase, β-glucosidase, α-amylase, fructosyltransferase), which make intracellular compounds more easily extractable [31]. The plant material is pretreated with enzymes to hydrolyze the cell walls and release the phytochemicals bound to lipid and carbohydrate chains inside the cell [32]. This is followed by solvent extraction, pressurized hot water extraction, or other green extractions, for the recovery of volatile compounds, hydrophilic and hydrophobic pigments, phenolic compounds, and other bioactive compounds [33]. The EAE process is cost-effective, easy, and completely green. The efficiency of the process depends on the type, dosage, and required condition of the enzymes, a time-temperature combination of the process, properties of the plant material such as particle size, water content, chemical composition, and solvent-to-solid ratio [34]. EAE has several benefits, such as high extraction yield and quality, green extraction (because of water and enzyme applications that are of natural origin), capability for scaling up the process, and reduced extract filtration and purification process [35]. The enzyme pretreatment of raw material normally results in a reduction in extraction time, minimizes usage of solvents, and provides increased yield and quality of product. Many published works showed how the enzymatic pre-treatment can improve the extraction yield of bioactive compounds such as polyphenols and oils from plant raw material [36,37,38]. Additionally, the complementation of EAE with bacterial cultures represents the new frontier for sustainable extraction of bioactive molecules from plant waste, further limiting the use of expensive commercial enzymes [35,39]. Among the used bacteria, *Bacillus subtilis* has gained high interest for the industrial production of several products, such as degradative enzymes, heterologous proteins, bio-insecticides, and antibiotics, all Generally Recognized as Safe (GRAS) bio-products [40,41]. Recent work has proposed the sustainable extraction of bioactive compounds from plant waste based on a combined EAE and low-cost pretreatment with *B. subtilis* [42]. A bacterial culture containing a wild-type (WT) and an overproducing cellulases and xylanases strain (OS58), was used on cauliflower (*Brassica oleracea*) waste materials. This pilot scale study showed that the recovery of polyphenols has been substantially enhanced by the pretreatment procedure. The developed process is highly sustainable as it includes the valorization of agri-food waste for the recovery of natural bioactive compounds with enhanced productivity and use of clean energy, thus following the principles of circular economy.

## 4. Why Use Biostimulants as Seed Treatments?

In the agricultural field, biostimulants have garnered significant attention as viable substitutes for chemical fertilizers, and they play a crucial role in organic farming and sustainable crop production management. Therefore, they can be considered as Nature-based solutions (NBS) applicable for agricultural systems. Biostimulants can be categorized into two main types according to their source. The first category comprises biostimulants obtained from natural sources, such as bacteria and plants. The second category encompasses non-biological sources, including physical factors and chemicals [43]. Initially, PBs have been used in organic agriculture mostly for high-value horticultural goods. However, in recent times, PBs have also been employed in traditional crop cultivation because of their potential to enhance plant productivity and quality, while still meeting economic and sustainability production standards [44]. The definition of PBs is mostly based on the response they provoke in plants rather than their composition, as this group encompasses a wide range of compounds such as sugars, amino acids, proteins, nucleic acids, polysaccharides, phenolic acids, and flavonoids. Considering the presence of such diverse bioactive compounds, PBs can affect phenotypic characteristics and increase crop yield through improved stress tolerance, nutrient absorption, and assimilation [45]. Particularly under adverse environmental conditions, PBs have been demonstrated to enhance leaf pigmentation, photosynthetic efficiency, leaf area and number, shoot and root biomass, fruit count, and/or average weight in many species [46,47,48]. Biostimulant activity is influenced by various factors such as plant genotype, growth conditions, dosage, and application time [45]. A recent review [49] emphasizes the significance of the PBs application method. Some of the important factors to be considered while choosing the relevant mode of action are: (i) the uptake of biostimulants after application, (ii) knowledge regarding the bioactive compounds, and (iii) the absorption capacity of the target organ. Taking these parameters into consideration can improve PBs effectiveness, resulting in better outcomes, particularly in terms of economic aspects.

Commonly known methods of application include soil preparations in which powders, granules, or solutions are added to the soil. One drawback of these treatments is that the physicochemical properties of the soil could interfere with the uptake. An alternative approach is to use foliar spraying so that the product is absorbed through the stomata. This approach is most effective during daylight hours when the stomata remain open, allowing for optimal absorption of nutrients. On the other hand, seed pretreatment could be considered as one of the most beneficial modes of application (Figure 3). Seed pretreatments in the form of seed priming can enhance nutrient absorption speed and rapidly activate the pre-germinative metabolism [50,51]. This activation of the pre-germinative metabolism involves essential metabolic pathways like cellular respiration, antioxidant mechanisms, and DNA damage response [52,53]. Seed priming is a pre-sowing technique in which seeds are subjected to controlled imbibition with water or other priming agents, and subsequently desiccated to return to their original moisture content. The advantages of seed priming have been widely recorded. Germination is improved and synchronized while abiotic and biotic stress tolerance is enhanced [6,54]. The application of PBs during this phase may further improve seed vigor and provide more nutrients and defense support in the form of bioactive molecules. Given the diversity of bioactive molecules present in PBs, the beneficial effect that plants experience may not be the result of the actions of individual components but rather of the synergistic interactions between multiple substances [55]. The next section summarizes examples of studies where different types of PBs have been applied as seed treatments, underlining their effect at a physiological or molecular level.

## 5. Examples of the Application of Plant-Based Biostimulants as Seeds Treatments

As we have reported in the previous sections, PBs can be sustainably obtained from agricultural by-products [56] and used as seed treatments. These can be produced as commercial formulations (e.g., Kelpak^®^, KIEM^®^) or studied as plant extracts, rich in numerous bioactive molecules (Table 1). Among the commercially available products, Kelpak^®^ is a seaweed-based biostimulant derived from *Ecklonia maxima* and shown to improve germination performances in okra (*Abelmoschus esculentus*) seeds [57], as well as drought and salinity stress in African foxglove (*Ceratotheca triloba*) [58]. KIEM^®^ is another commercial product, derived from lignin sources. Its application in cucumber (*Cucumis sativus*) and soybean (*Glycine max*) resulted in increased germination and seedling biomass, decreased H_2_O_2_ content, and accumulation of gene transcripts encoding for scavenger enzymes under heat stress [59,60].

Microalgae are versatile sources for applications in pharmaceuticals, nutraceuticals, food and feed production, aquaculture, bioenergy, and environmental initiatives. Algal polysaccharides exhibit biostimulant properties that can enhance plant growth, nutrient uptake, and tolerance to both biotic and abiotic stresses [61]. Extracts from the microalga *Acutodesmus dimorphus* have resulted in faster germination and seedling growth in tomatoes (*Solanum lycopersicum*) [62]. Similar results were obtained when *Scenedesmus obliquus* extracts were applied to garden cress *(Lepidium sativum*) [63]. *Chlorella vulgaris* extracts enhanced germination percentage and speed in tomato and barley (*Hordeum vulgare*) [64]. Presoaking and irrigation with aqueous extracts from the brown alga *Sargassum polycystum* applied on *Vicia faba* and *Helianthus annuus* seeds, has improved germination and increased the phenolic and flavonoid contents [65]. Extracts from the green alga *Cladophora glomerata* applied to lupin (*Lupinus angustifolius*) seeds, resulted in enhanced germination and seedling growth [66].

The category of plant-based extracts includes multiple promising natural-based solutions (NBS) for the development of next-generation bio-products suitable for sustainable agricultural practices [67]. When considering seed treatments, moringa (*Moringa oleifera*) leaf extracts have been shown to improve bell pepper (*Capsicum annuum*) seed germination and growth under heavy metal and salinity stress [68]. Another example includes the application of *Cuscuta reflexa* extract (CRE) on wheat seeds, which at low doses, was able to mitigate the adverse effects of water stress on seed germination parameters by stimulating the activity of multiple enzymes (proteases, amylases, glucosidase) [69]. Additionally, the application of raw and fermented alfalfa (*Medicago sativa*) brown juice on French marigold (*Tagetes patula*) seeds also resulted in improved germination and seedling growth [70]. Other aqueous extracts that have been tested on different plants confirmed multiple times the positive effects of these types of seed treatments on improving germination performance. These include *Posidonia oceanica* extracts applied on tomato [71], *Atriplex halimus* extract applied on sorghum (*Sorghum bicolor*) [72], and sunflower extracts applied to peas (*Pisum sativum*) [73].

All these products are rich in bioactive compounds which exercise their beneficial roles through antioxidant and antimicrobial activities [74]. Among these, flavonoids are polyphenolic compounds produced in plants as secondary metabolites; they play a crucial role in plant growth and development under severe environmental limitations. Bioflavonoids extracted from *Citrus* spp. fruits were shown to improve the germination of canola and soybean seeds under salinity stress [75]. Melatonin (MEL) is another multifunctional biomolecule known to be involved in several physiological processes, from seed germination to post-harvest fruit storage and seed longevity [76]. It has been recently detected in various monocotyledonous and dicotyledonous plants, and its use as a priming agent has led to improved seed germination, alleviation of salinity stress, and up-regulation of steviol glycosides gene expression in *Stevia rebaudiana* seeds [77]. When applied to maize seeds, it improved germination during cold stress by promoting antioxidant defense [78]. Biochemicals from garlic have been previously shown to enhance crop production in pepper, eggplant, and cucumber [79]. Garlic allelochemicals also improved seed germination and salinity stress tolerance in tomatoes [13]. Plant protein hydrolysates (PPHs) are another group of biomolecules obtained from raw materials like soybean, rice, wheat, and maize, and they are characterized as refined protein forms containing peptides, oligopeptides, and amino acids [80]. A byproduct of the corn starch industry, CSL (corn steep liquor) protein hydrolysate has been shown to significantly increase wheat germination [81]. Thymoquinone (TQ) is a primary bioactive component from black seed oil (*Nigella sativa*) [82], reported to increase lentil germination and seedling vigor indexes along with alleviation of cadmium stress [83]. Other thyme essential oils were used to coat durum wheat seeds, resulting in enhanced germination and tolerance to drought stress [84]. Another use of plant bioactive compounds includes the sustainable production of nanoparticles (NPs) [85]. Saponins from quinoa extracts were used as reducing agents for silver ions to synthesize silver NPs, shown to promote radish (*Raphanus sativus*) seed germination [86]. Moreover, lignin NPs derived from alkali lignin have been reported to positively affect maize seed germination [87].

**Table 1 plants-13-01004-t001:** List of biostimulants used for seed treatments. The table includes types of products, plant-based sources, species to which these were applied, and effects obtained in terms of germination, along with pertinent references.

Type	Product	Source	Application on Seeds	Advantages	References
Commercial	Kelpak^®^	*Ecklonia maxima*	*Abelmoschus esculentus*	Improved germination indexes (FGP, GI, GRI)	[57]
			*Ceratotheca triloba*	Improved germination under drought and salinity stress	[58]
	KIEM^®^	Lignins, amino acids, molybdenum	*Cucumis sativus* *Glycine max*	Increased germination and biomass, improved antioxidant defense under heat stress	[59,60]
Algae	Microalgae extracts	*Acutodesmus dimorphus*	*Solanum lycopersicum*	Enhanced germination and plant growth	[62]
		*Scenedesmus obliquus*	*Lepidium sativum*	Increased germination index	[63]
		*Chlorella vulgaris*	*Solanum lycopersicum* *Hordeum vulgare*	Increased germination percentage and germination index, decreased germination time	[64]
	Brown alga	*Sargassum polycystum*	*Vicia faba* *Helianthus annuus*	Improved SVI, radicle and seedling length, fresh weight, increase in phenolic and flavonoid contents and the total antioxidant capacity	[65]
	Green alga	*Cladophora glomerata*	*Lupinus angustifolius*	Enhanced germination percentage, root, hypocotyl and epicotyl length	[66]
Plant Extracts	Leaf extracts	*Moringa oleifera*	*Capsicum annuum*	Improved germination and seedling growth under heavy metal and salinity stress	[68]
		*Atriplex halimus*	*Sorghum bicolor*	Cadmium stress alleviation Enhanced GP, SVI, decreased MGT, stimulated antioxidant enzymes	[72]
	Fresh biomass	*Cuscuta reflexa*	*Triticum aestivum*	Water stress mitigation	[69]
	Brown juice	*Medicago sativa*	*Tagetes patula*	Increased germination parameters, seedling growth and biomass	[70]
	Dry biomass	*Posidonia oceanica*	*Solanum lycopersicum*	Increased germination index and root length	[71]
		*Helianthus annuus*	*Pisum sativum*	Boosted SG, GI and VI	[73]
Bioactive Compouds	Bioflavonoids	*Citrus* fruits	*Brassica napus Glycine max*	Increased germination under salinity stress	[75]
	Melatonin (MEL)	Monocot and dicot families	*Stevia rebaudiana*	Improved germination and salinity stress alleviation	[77]
			*Zea mays*	Improved germination and embryonic axis growth during chilling stress	[78]
	Garlic extracts	*Allium sativum*	*Solanum lycopersicum*	Improvement of seed germination and alleviation of salinity stress	[13]
	CSL protein hydrolysate	*Zea mays*	*Triticum aestivum*	Increase in germination parameters	[81]
	Thymoquinone	*Nigella sativa*	*Lens culinaris*	Increased germination indexes and alleviation of cadmium stress	[83]
	Thyme essential oil	*Thymbra capitata*	*Triticum turgidum*	Enhanced germination and drought tolerance	[84]
	Saponins	*Chenopodium quinoa*	*Raphanus sativus*	Improved germination indexes with no phytotoxicity	[86]
	Lignin nanoparticles	Alkali lignin	*Zea mays*	Increased germination, radicle length, fresh weight, shoots and roots length	[87]
Plant Waste Material	Hydrochar	*Saccharum officinarum*	*Zea mays*	Increased germination rate	[88]
	Spent coffee grounds	*Coffea arabica*	*Brassica* spp.	Improvement of germination	[89]
	Compost material	Artichoke *(Cynara cardunculus var. scolymus)*	*Zea mays*	Increased germination rate, primary and lateral root length	[90]

Other sources of PBs derived from plant waste material include hydrochar, spent coffee grounds, and compost. Hydrochar obtained from the hydrocarbonization process of by-products from the sugarcane industry (bagasse, vinasse and a mixture of both), was shown to increase the germination rate of *Z. mays* seeds [88]. Spent coffee grounds have been used as biostimulants for seed germination in various Brassicaceae (cauliflower, broccoli, cabbage) [89]. Artichoke water extractable organic matter (CYN-WEOM) has been applied to increase germination rate, as well as primary and lateral root length, in maize seedlings [90].

All of these examples indicate that bioactive molecules extracted from plants and plant wastes can be easily repurposed for the support of the seed industry. Most studies focus on phenotypic characterizations, so showing improved germination and seedlings growth, but also stress tolerance. Few studies have looked at some molecular mechanisms, and these mostly refer to enhanced antioxidant, enzymatic, or non-enzymatic mechanisms. The high variability of molecules, as well as sources to extract them from, point towards a huge reservoir of solutions to tap into in the search for sustainable, nature-based solutions for the future of agriculture.

## 6. Conclusions

To conclude, the present review of the literature straightforwardly indicates that the integration of PBs as seed priming agents is a promising approach to enhancing agricultural sustainability and productivity within the framework of the circular economy. This approach aligns with the principles of reducing waste, reusing resources, and recycling nutrients, thus contributing significantly to the consolidation of a more sustainable and resilient agricultural sector. Numerous advantages can be attributed to the use of these nature-based solutions, while further studies need to be continuously designed to overcome specific challenges as well. To cite some of the advantages or strengths of this approach, we can say that the application of PBs on seeds can: (1) enhance seed vigor and germination, leading to healthier and more robust crops, a crucial attribute for ensuring food security and maximizing agricultural outputs; (2) reduce the use of synthetic chemical inputs, minimizing the environmental footprint of agricultural practices; (3) improve resource efficiency in terms of both plant nutrition as well as transforming waste materials in valuable inputs; (4) boost soil health by enhancing microbial activity and soil fertility; (5) bring overall economic benefits in terms of cost savings for several stakeholders, including farmers. Nonetheless, several challenges remain to still be addressed, including: (i) the consistency and standardization of treatment applications, given the variability in the composition of PBs; (ii) regulatory hurdles related to the lack of clear regulatory frameworks for the approval and use of PBs across different regions, thus still hindering their widespread adoption and market development; (iii) the need for better awareness and understanding among farmers and agronomists related to the benefits and application of PBs; (iv) the need for further research and development programs to elucidate their mechanisms of action, optimize formulations, and tailor applications to specific crop and environmental conditions; (v) the initial costs and the economic viability of integrating PBs into existing agricultural practices can be a barrier for some farmers, particularly in developing countries. As a take-away message, it should be underlined that the journey towards fully harnessing the potential of PBs in agriculture requires collaborative efforts among scientists, industry stakeholders, policymakers, and the farming community. Embracing innovation, fostering research and development, and navigating regulatory landscapes are essential steps in overcoming the existing challenges, moving towards advancing agricultural sustainability, bolstering food security, and embodying the principles of a circular economy.

## Figures and Tables

**Figure 1 plants-13-01004-f001:**
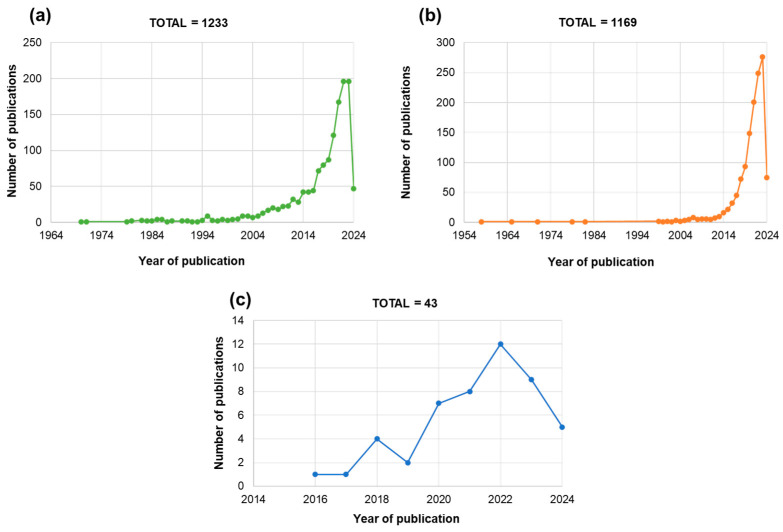
Trends in article publications based on data available in the PubMed database on 21 March 2024. Specific keyword combinations were used as follows: (**a**) “Seed Priming” NOT “Review”. (**b**) “Plant Biostimulants” NOT “Review”. (**c**) “Seed Priming” AND “Plant Biostimulants”.

**Figure 2 plants-13-01004-f002:**
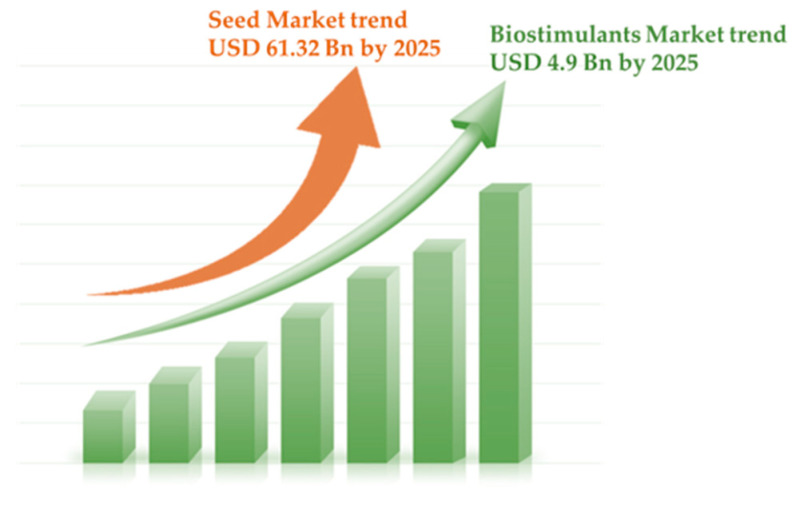
Global seed and biostimulants market trends predictions for 2025. Values were reported in Fortune Business Insights for the Seed market (https://www.fortunebusinessinsights.com/industry-reports/commercial-seed-market-100078, accessed on 29 March 2024) and Markets and Markets for plant biostimulant market (https://bit.ly/3tgd4D3, accessed on 29 March 2024).

**Figure 3 plants-13-01004-f003:**
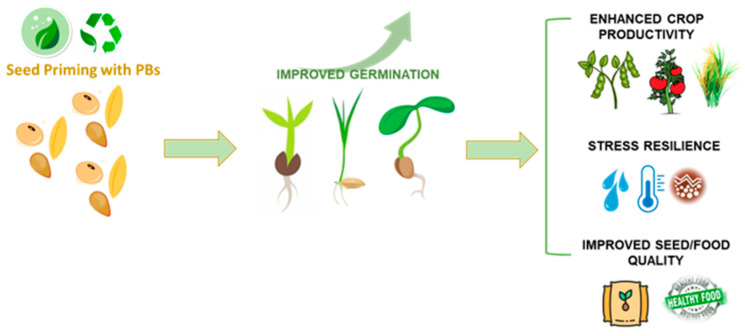
Schematic representation underlining the benefits of PBs (obtained from recycled agri-food waste) applied as seed priming treatments in different crops. The positive effects span from improved germination to enhanced productivity, stress resilience, and final seed/food quality.

## Data Availability

No new data were created or analyzed in this study. Data sharing is not applicable to this article.

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
