# Peer review of "Plant-Based Biostimulants for Seeds in the Context of Circular Economy and Sustainability"

_plants, 2024, doi:10.3390/plants13071004_

Round 1

Reviewer 1 Report

Comments and Suggestions for Authors

The work does not present any scientific advance in terms of the subject covered. However, I consider the topic interesting and I believe that pertinent information has been gathered on the subject.

It is another work on this topic and although it is a review work, it does not correspond to a systematic review on the topic, not explaining, for example, why this bibliography was chosen and not another of the more than 1000 existing references.

Things to consider:

1. The abstract does not present any conclusion on the topic, leaving the abstract as the weakest part of the work. I recommend an in-depth review of this summary.

2. The development of the work is adequate, although tedious to read. If possible, it should be more targeted and incisive. The work seems to me to be too extensive for its approach to the theme.

3. The conclusions are very vague and largely repeat the introduction, and should be rewritten in a more concise and more focused on the work

Author Response

Reviewer 1

The work does not present any scientific advance in terms of the subject covered. However, I consider the topic interesting and I believe that pertinent information has been gathered on the subject.

It is another work on this topic and although it is a review work, it does not correspond to a systematic review on the topic, not explaining, for example, why this bibliography was chosen and not another of the more than 1000 existing references.

Things to consider:

  1. The abstract does not present any conclusion on the topic, leaving the abstract as the weakest part of the work. I recommend an in-depth review of this summary.
  2. The development of the work is adequate, although tedious to read. If possible, it should be more targeted and incisive. The work seems to me to be too extensive for its approach to the theme.
  3. The conclusions are very vague and largely repeat the introduction, and should be rewritten in a more concise and more focused on the work

REPLY: Authors thank the Reviewer for the appreciation of the work and insightful suggestions to improve it. In the revised version, we carefully addressed improving abstracts and conclusions as well as removing repetitions and being more straightforward with the different findings. To better explain the choice of the topic and references, an additional graph was added in Fig. 1 showing that even though the two topics, when looked separately, include a considerable amount of articles published, if we combined them in a unique search it is possible to evidence a pronounced gap of knowledge regarding the use of biostimulants as seed priming agents. In this case, the PubMed search using “Seed Priming” AND “Plant Biostimulants” as keywords, evidenced only 43 articles of recent publication during 2016-2024 (Fig. 1c) and we based our work mainly on these publications, while choosing only specific ones (limited in number) for the single PBs or seed priming topics. In the revised version, all the changes brought to the manuscript are presented in dark red for the ease of following them during the next step in the revision process. We hope the Reviewer will find this revised version more suitable for publication.     

Reviewer 2 Report

Comments and Suggestions for Authors

Overall, this is well organized and clearly composed, I have only a few comments and/or corrections for clarification of some sentences, etc. for the authors to consider.

Line.                Comment and suggested revision of sentences.

12        --- development of seed-specific plant-based biostimulants ----

46        --- Efficient seed germination under adverse ---

50        These consist of adaptable, ----

52        Seed priming has been shown -----

54        --- in a wide range of environments [7-9].

64        --- reducing synthetic agrochemical inputs.

78-79   ----- phytohormones, peptides, and minerals [15].

104      ----challenges; on the one hand, they must be ----

105      --- on the other hand, they must address ----

124-125    ---- not only as a means to start ----

183      -----reducing energy costs; and, at the same time, increasing ----

194      ----  reasons, often the resultant processing cost may be high,  with low -----

209      One of the current major research challenges is the ----

217      ---- each of these processes must have a reduced -----

220      ---macroscopic and microscopic characteristics of the ---

221      ----purify; the first ones are referred to----

222      ---- while the second ones are related ----

232      ---- systems are more widely used, depending on ----

244      ----- appropriate, and preserve the functional -----

254      ----- by hydrotropic solvents or amphiphilic organic salts ---

265      --- allowing the precipitation of the ---

300      ----- cultures represents the new ---

301      ---- waste; thus, limiting the use ---

351      ---- and activate the pre-germination processes that trigger radicle---

363      The next subsection summarizes ----

371      As we have reported in the previous subsections, plant-based---

373      In the perspective of sustainability ----

379      These encompass a large variety of sources spanning ---

388      ---- resulted in increased germination ---

397      ---- have resulted in faster ---

421      simply for consistency consider adding the formal taxonomic name of tomato

                        ---root length of tomato (Solanum lycopsersicum) [72].

440      ---allelopathic properties that have been shown to influence numerous aspects of crop production, including---.

469-471           ---- spent coffee grounds and compost. Hydrochar, obtained from the hydrocarbonization process of by-products from the sugarcane industry (bagasse, vinasse and a mixture of both), was shown to  increase the germination rate of Z. mays seeds [88].

495      ----priming is envisioned as a sustainable, economic ----

497      ----is a well-known seed quality improvement technique, demonstrated to ---

504      ---- its own set of issues, and one of the challenging aspects of this field is to find----

511  Suggestion to authors: use the word ‘subsection’ not chapter. This is not a book; it is  an interesting published paper with subsections.

Comments on the Quality of English Language

Overall, the manuscript is well composed and clearly organized, I have made a few suggestions to the authors for clarification of English usage, etc.

Author Response

REPLY: Authors are thankful to the Reviewer for the appreciation of the work and suggestions given to improve it. All the indications were followed and errors solved. In some cases, sentences were better formulated or removed to avoid repetitions. The changes made in the manuscript are presented in dark red for the ease of tracking during the next step in the revision process.

Reviewer 3 Report

Comments and Suggestions for Authors

Dear Authors,

thank you for the opportunity to meet the review entitled: "Plant-based Biostimulants for Seeds in the Context of Circular Economy and Sustainability".

It can be concluded that PBs are now increasingly being implemented across different cropping systems. In addition, the "Green deal" or various other ecological strategies increase the potential for application in the future as well. As mentioned in this article, there is a lot of research that pays attention to the impact of PBs on the yield, physiology, or quality of plants. However, there are also numerous reviews that describe these effects from different perspectives. There is therefore little space to provide new, groundbreaking knowledge in this area.

The authors succeeded in this. This review provides a comprehensive view of the PBs application using seed priming. The individual chapters are in a logical sequence and adequate proportionality. The review is supplemented with appropriate graphic elements that increase its quality. Literary sources are suitable, up-to-date and in sufficient number.

Author Response

REPLY: Authors are very thankful to the Reviewer for the appreciation of the work. Following suggestions coming from multiple Reviewers, the manuscript was revised to further improve its quality. The changes made in the manuscript are given in dark red for easier tracking.  

Round 2

Reviewer 1 Report

Comments and Suggestions for Authors

The abstract is still too long, although it has improved substantially. The first 6 lines of the abstract are essentially an introduction to the topic, which seems too much to me.

Other databases should have been used to carry out the bibliographic review, although the bibliography used is mostly relevant and comprehensive.

The added bibliographic reference (lines 561-562) does not respect the format used by this journal

Author Response

Part 2

The abstract is still too long, although it has improved substantially. The first 6 lines of the abstract are essentially an introduction to the topic, which seems too much to me.

REPLY: We thank the Reviewer for his/her thorough evaluation of the revised work. We have now further improved the abstract to avoid long introductory aspects and diving more directly into PBs and the importance of their use for seed treatments. We have also shortened and better defined the structure and results that emerged from consulting this specific topic. Hopefully, you can find that the quality of the abstract is further improved.   

Other databases should have been used to carry out the bibliographic review, although the bibliography used is mostly relevant and comprehensive.

REPLY: We agree with the Reviewer that other databases, such as Scopus or Web of Science, can be explored to have a better, more integrative view any topic; although, PubMed is recognized as a versatile, well-organized, user-friendly, and prolific collection of peer-reviewed articles within the scientific community. In this work, the point of Fig. 1 was not to do a complete meta-analysis of databases and extract or extrapolate the relevance of discussed topics, but rather to underline that few has been done and there is space for much more in the future; at least for the seed priming with biostimulants approach. This could also stimulate other works that can be dedicated to specific meta-analyses approaches regarding biostimulants in the near future.    

The added bibliographic reference (lines 561-562) does not respect the format used by this journal

REPLY: The indicated references were corrected to follow the format of the journal.
